# Effects of Biological Age on Athletic Adaptations to Combined Plyometric and Sprint with Change of Direction with Ball Training in Youth Soccer Players

**DOI:** 10.3390/biology12010120

**Published:** 2023-01-12

**Authors:** Hamza Marzouki, Samar Sbai, Ibrahim Ouergui, Okba Selmi, Marilia S. Andrade, Ezdine Bouhlel, Mabliny Thuany, Katja Weiss, Pantelis T. Nikolaidis, Beat Knechtle

**Affiliations:** 1High Institute of Sports and Physical Education of Kef, University of Jendouba, Kef 7100, Tunisia; 2Departamento de Fisiologia, Universidade Federal de São Paulo (UNIFESP), São Paulo 04021-001, SP, Brazil; 3Laboratory of Cardio-Circulatory, Respiratory, Metabolic and Hormonal Adaptations to Muscular Exercise, Faculty of Medicine Ibn El Jazzar, University of Sousse, Sousse 4000, Tunisia; 4CIFI2D, Faculty of Sports, University of Porto, 4200-450 Porto, Portugal; 5Institute of Primary Care, University of Zurich, 8006 Zurich, Switzerland; 6School of Health and Caring Sciences, University of West Attica, 12243 Athens, Greece; 7Medbase St. Gallen Am Vadianplatz, 9000 St. Gallen, Switzerland

**Keywords:** team sports, testing, strength conditioning, peak height velocity, physical fitness, in-season period

## Abstract

**Simple Summary:**

The combination of plyometric and sprint with change of direction (COD) training was shown to be more attractive and effective in eliciting greater developments in youth soccer fitness than a single training regime (e.g., plyometric protocol). However, the evidence is not well-known regarding the in-season effects of biological age (peak height velocity—PHV) on motor adaptive processes following combined plyometric and COD with ball training (P-CODBT) in youth soccer players. Thus, this study aimed to examine the effects of a short combined plyometric and CODB training protocol (8 weeks and twice a week) on some measures of athletic performances of male youth soccer players, circa- and post-PHV, within the competitive season. The data showed that the experimental condition induced significant additional gains only on explosive measures compared to the control condition. The data showed that a short-term P-CODB (two sessions per week) can be a safe strategy and induce meaningful benefits in the explosive performances (i.e., speed with and without a ball, jump, and COD with and without a ball) of youth soccer players during the season period. Therefore, coaches and practitioners can integrate the protocols proposed in this study into their training weekly routines as part of well-structured warm-up before specific soccer training. Since training programs should incorporate drills that closely resemble competition motions, the use of P-CODBT may be a suitable and alternative strategy to improve both the athletic and technical abilities of youth soccer players. However, the improvements generated by P-CODBT were not affected by biological age.

**Abstract:**

There is evidence for the effectiveness of youth combined plyometric and sprint with change of direction (COD) training. However, the evidence is not well-known regarding the in-season effects of biological age (peak height velocity—PHV) on the motor adaptive processes following combined plyometric and COD with ball training (P-CODBT) in youth soccer players. This study aimed to examine the in-season effects of P-CODBT (8 weeks and twice a week) on the athletic performances of male youth soccer players, circa- and post-PHV. In a randomized controlled training study with pre-to-post measurements, forty-eight male players were assigned into two experimental (performing P-CODBT; *n* = 12 × circa-PHV and *n* = 12 × post-PHV) and two control groups (CONG; *n* = 12 × circa-PHV and *n* = 12 × post-PHV). The pre- and post-training participants were assessed for their anthropometric, linear sprinting with and without a ball, COD speed with and without a ball, vertical jump, dynamic balance, and endurance-intensive performances. After the intervention, the experimental condition induced significant (all *p* < 0.0001) and small to large effect size (ES = 0.263–3.471) additional gains only on explosive measures compared to CONG. Both the experimental (all *p* < 0.0001; ES = 0.338–1.908) and control (*p* = 0.011–0.0001; ES = 0.2–1.8) groups improved their athletic performances over the training period. The improvements generated by *p*-CODBT were not affected by biological age. In-season short-term P-CODBT (twice a week) could be safe way to generate benefits in explosive performances in youth soccer players, which are relevant components of match-winning actions in soccer.

## 1. Introduction

Soccer is an intermittent-intensity sport that involves both the aerobic and anaerobic metabolism. Mental, physical, technical, and tactical skills are required for a player to succeed in a competition [1,2]. During competitive matches, time and motion analyses have shown that players complete 1300 changes of activity [3], including 220 high-intensity movements over short distances (i.e., acceleration/decelerations, sprints, jumps, and changes of direction (COD)) [4], which are associated with specific actions (i.e., tackling, defending, heading, and creating space during possession), in sprints, with or without a ball, being the most common action before crucial moments (e.g., scoring or conceding a goal or winning the ball) [5,6,7,8].

Although sprinting with the ball represents only 1.2–4.6% of the total distance covered by players [9], the physiological stress could be higher when running with the ball compared to normal running [10], and physical actions with the ball could be affected by match-induced fatigue [11]. For a given locomotion’s speed, Reiley [12] has mentioned that the training stimulus was higher when players were running with the ball compared to normal running, therefore suggesting the positive effects of using soccer-specific routines wherever possible. As a result, various training modalities based on dribbling actions were designed to develop the physical performances (e.g., running speed, COD ability, jumping, agility, and intermittent endurance (maximal oxygen consumption—VO2max)) in soccer players at different competitive levels and ages, such as circuits with the ball and small-sided games [10,13,14,15]. Recently, it has been shown that COD with ball training (CODBT) is effective in conditioning soccer-related fitness among youth players [13]. Given the frequent in-game situations that allow for a sudden COD action [16], it would be plausible to plan CODBT in conjunction with other training modalities (e.g., plyometrics) in order to optimize physical fitness as well as some technical (e.g., dribbling ability) and tactical (e.g., counterattacking) skills in youth players [13].

As previously documented, soccer players have to perform a wide range of motions (e.g., accelerations, turns, and headings) that require a high-power development output by the lower limb muscles, and play a crucial role in the match outcome [17,18]. Plyometric training (PT) is an effective way to promote components of muscle power (e.g., acceleration, explosive strength, speed, and jumping ability) in youth soccer players [17,19]. Such a training strategy extensively stresses the stretch-shortening cycle (SSC) muscle action during jumping exercises [19]. The SSC is of great importance in PT based on its participation in enhancing the muscle–tendon unit’s ability and producing the maximal force in the shortest time [20]. The combination of this kind of training with CODBT may be more attractive and effective in eliciting greater developments in youth soccer fitness compared to a single training regime (e.g., plyometric protocol).

Despite today’s compelling evidence for the effectiveness of youth combined plyometric and sprint with change of direction training [1,17], the evidence is not well-known regarding the in-season effects of different moderator variables, such as biological age (i.e., peak height velocity—PHV), on motor adaptive processes following combined plyometric and CODB training programs (P-CODBT) for young soccer players. From puberty, it is expected that a marked gain of muscle mass and strength, especially among the male athletes, is a consequence of the maturity of the neuroendocrine system [21]. This could be of interest, given the debate on the timing of youth training and the potential existence of “golden periods” of adaptation [22]. In fact, a prior meta-analysis [23,24,25] and some randomized controlled interventions [26,27] regarding maturation-related athletic adaptations to various modes of training (e.g., PT and combined resistance training) in male youth athletes indicated conflicting results between more- and less-mature athletes. However, to the best of our knowledge, only one interventional study has examined the influence of biological age (circa- vs. post-PHV) on PT-related effects on the athletic performances of male youth soccer players [26]. In the aforementioned study, no significant differences in motor performance gains were found across the circa- and post-PHV groups. Addressing this shortfall in knowledge may be highly important within the context of P-CODBT training prescription for male youth soccer players of different biological ages. Thus, this study aimed to examine the effects of a short combined plyometric and CODB training protocol (8 weeks and twice a week) on some measures of athletic performances (i.e., short linear sprinting with and without a ball, COD speed with and without a ball, jumping, dynamic balance, and endurance-intensive performances) in male youth soccer players, circa- and post-PHV, within the competitive season. We hypothesized that the experimental condition induced significant additional gains on athletic measures compared to the control condition, with no biological age-related differences.

## 2. Materials and Methods

### 2.1. Participants

A statistical software (G*Power software, version 3.1.9.4, University of Kiel, Kiel, Germany) was used to calculate the sample size. Given the study design (4 groups × 2 times), the effect sizes considered to generate the sample size estimation were attained based on tabled data from previous research [19,27,28]. The analysis resulted in a minimum of 8 participants required in each group (effect size f = 0.25, α = 0.05, and 1 − ß = 0.80; actual power = 82.80%). Thus, forty-eight healthy male youth players from the same regional level soccer team (under 15 and under 17) and with similar training habits volunteered to participate in this study. Participants were categorized into 2 maturity groups following a previously described equation by Mirwald et al. [29]. The Mirwald method is a simple, noninvasive, and inexpensive method of predicting years from PHV as a measure of maturity offset using anthropometric measurements. Briefly, maturity offset = −9.236 + (0.0002708 × leg length × sitting height) + (−0.001663 × age × leg length) + (0.007216 × age × sitting height) + (0.02292 × body mass/body height ratio) [29]. Based on PHV offset, subjects can be classified into 3 categories: pre-PHV (−3 to >−1 years), circa-PHV (−1 to +1 year), and post-PHV (>1 to +3 years) [29]. Random allocation was maintained using the method of randomly permuted blocks stratified by biological age, which resulted in the following assignments: 2 experimental (*n* = 12 × circa-PHV and *n* = 12 × post-PHV) and 2 control (CONG; *n* = 12 × circa-PHV and *n* = 12 × post-PHV) groups. To participate in the study, the inclusion criteria were as follows: (a) to be safe from musculoskeletal injuries for one year before the study, (b) to have at least 2 years of soccer experience and participate regularly in the club training routines, (c) to have accomplish ≥95% of training sessions, and (d) to not have any background in formalized strength and power training and have no prior experience with P-CODBT programs. Participants competed regularly at regional championship. Prior to the study, parental informed consent was obtained as participants were younger than 18 years. Participant characteristics per biological age and condition are illustrated in Table 1. The local research ethics committee approved the protocol (approval number: 011/2021; date of approval: 19 September 2021), in the spirit of the Helsinki Declaration [30].

### 2.2. Experimental Design

The current study adopted a randomized controlled trial with pre-to-post measurement design to examine the effects of 8 weeks of P-CODBT on athletic performances of male youth soccer players, circa- and post-PHV. Participants were assigned according to their biological age into 2 experimental groups (performing P-CODBT twice per week) and 2 control groups. The experimental groups replaced a part of their standard technical–tactical drills with P-CODB training (Tuesday and Thursday), while the CONG completed skill-development activities according to their standard training routines, during the interventions across all the training study to satisfy the randomized controlled design assumed. The usual micro-cycle training for all cohorts consists of 5 sessions a week (~70 to 90 min each session), with a weekly match scheduled on Sunday (Table 2). Briefly, during the first three days of the micro-cycle training, ~60% of the time was devoted to technical–tactical tasks and ~40% to physical training, while the last two days devoted ~75% of the total time to technical–tactical tasks and remaining time to physical training (Table 2).

The experimentation was carried out within the competitive season (from January to March), lasted 10 weeks, and consisted of 2 testing weeks (pre and post-test) and 8 weeks of specific training. One week before the beginning of the experimentation, all participants participated in two separate familiarization sessions (~48 h) to ensure their technical proficiency of performing testing and training procedures. At pre- and post-testing, all participants were assessed for anthropometric, linear sprinting with and without ball, COD speed with and without ball, vertical jump, dynamic balance, and endurance-intensive performances by the same evaluators who were not blinded to groups. The reliability (i.e., intraclass correlation coefficient (ICC)) was calculated using the results of the second familiarization session and pre-test.

All tests were carried out ≥48 h after the most recent weekly match or training session to reduce the fatigue influence, and under similar standardized conditions (e.g., on a synthetic grass soccer field) and verbal encouragement over 2 non-consecutive days 48 h apart (Tuesday and Thursday evening from 18:00 to 20:30 h). The first testing day was devoted to assessing anthropometric parameters (body weight was measured to the nearest 0.1 kg using a digital scale (OHAUS, Florhman Park, NJ, USA), body height, leg length, and sitting height were measured to the nearest 0.01, and body mass index (BMI) was determined (kg·m^−2^)), linear sprinting with and without ball, and COD speed with and without ball performances. The second day was devoted to assessing vertical jump, dynamic balance, and endurance-intensive performances. Each testing session was preceded by a general 15 min warm-up consisting of jogging, active stretching, and sets of intense exercises (e.g., short sprints and skipping). With the exception of the endurance-intensive test, which was performed only once, the other tests involved two valid maximal repetitions separated by two minutes of passive recovery, and the best performance was retained for analysis. We allowed 5 min of passive recovery between tests.

### 2.3. Procedures

#### 2.3.1. Assessment of Linear Sprinting with and without Ball

To assess linear sprinting, the participants covered a 20 m distance with and without the ball as fast as possible, with times over 10 and 20 m (S10, S20, Ball-S10, and Ball-S20) recorded using a series of paired photocells (Globus, Microgate, Bolsano, Italy). Participants started from a standing position with the front foot placed 0.2 m behind the first photocell beam. The ICCs were 0.875 for S10, 0.965 for S20, 0.963 for Ball-S10, and 0.959 for Ball-S20, respectively.

#### 2.3.2. Assessment of Change of Direction Speed with and without Ball

Participants’ COD abilities were evaluated with the 15 m agility run with and without the ball (COD-15 m and Ball-15 m, respectively) as described by Mujika et al. [31]. In the 15 m agility run without the ball, participants were instructed: (a) to start sprinting 3 m behind the first photocell beam (start line); (b) to follow up with a 3 m sprint, then enter a 3 m slalom section marked by three sticks 1.6 m high and placed 1.5 m apart; (c) to clear a 0.5 m hurdle placed 2 m beyond the third stick; and to sprint 7 m to the second photocell beam (finish line).

The 15 m agility run with the ball was similar to the one without the ball, but participants were instructed to kick the ball under the hurdle while they cleared it. The participants then kicked the ball towards either of two small goals placed diagonally 7 m on the left and the right sides of the hurdle and then sprinted to the finish line. The ICC for COD-15 m and Ball-15 m were 0.812 and 0.895, respectively.

#### 2.3.3. Assessment of Vertical Jump

We adopted the counter movement jump test (CMJ) without arm swing to assess the jumping height (cm), according to Bosco et al. [32], using an infrared jump system (Optojump; Microgate, Bolzano, Italy). Participants had to start in a standing position, squat with their legs bent at 90 degrees, and immediately jump vertically as high as possible. Participants were required to place their hands on their hips to avoid the swinging effect of the arms on the test. Knees and ankles should be extended on takeoff, as well as on tiptoe landing. The ICC for CMJ was 0.986.

#### 2.3.4. Assessment of Dynamic Balance

We adopted the Y-balance test to evaluate the dynamic balance according to Chaouachi et al. [33]. The composite score was calculated and retained for analysis. The ICC for Y-balance test was 0.985.

#### 2.3.5. Assessment of Endurance-Intensive Performance

To assess the maximal aerobic velocity (MAV, km·h^−1^) and estimate the maximal oxygen consumption (VO_2_max) we used the multistage 20 m run test [34]. Between 2 parallel lines spaced 20 m apart, the participants were instructed to perform the greatest number of shuttles while respecting a running rhythm (dictated by a pre-recorded soundtrack). The initial speed of the test was 8 km·h^−1^ and it increased by 0.5 km·h^−1^ every minute. The test was stopped when the participants were no longer able to follow the imposed rhythm and could not reach the line in time when the beep sounded or stopped voluntarily. The ICC for MAV was 0.898. The VO_2_max was estimated using the following equation [34]:VO_2_max (mL·kg^−1^·min^−1^) = 31.025 + (3.238 × MAV) − (3.248 × A) + (0.1536 × MAV × A);
where A = age (years).

### 2.4. Plyometric Training

The training program was adapted from previous research [13,35,36] and consisted of four workshops, as visualized and quantified in Figure 1. In each workshop, participants began with plyometric drills (bouncy strides, hurdle jumps, single leg alternating hop jumps, and lateral hurdle jumps) and finished with a preplanned COD with ball. The training load progressively increased during the 8-week period (i.e., by increasing the set number) to minimize the risk of the occurrence of possible injuries. The characteristics of plyometric (i.e., volume and intensity) and COD (i.e., angles) exercises were based on prior recommendations [1,37]. A total of 90 s of passive recovery was allowed between repetitions and workshops. Participants were required to execute all drills at maximal intensity to ensure optimal training effects. No participants were excluded from the study due to injury.

### 2.5. Statistical Analysis

Descriptive data were presented as means ± standard deviation (SD). ICC was used to assess the relative reliability of each outcome measure. Shapiro–Wilk and Levene’s tests were used to test the distribution’s normality and homogeneity of variance for all variables before inferential statistics. Paired *t*-test examined the effectiveness of the training program within all groups. Percentage changes (Δ) were calculated for all athletic measures. Independent sample *t*-test was performed to examine differences between groups at pre-test. Regarding baseline differences in maturational groups, ANCOVAs (biological age (circa-PHV and post-PHV) × condition (experimental and CONG)) with pre-test as covariate were performed to test biological age and condition effects on the dependent variable values. Bonferroni post-hoc analyses were performed to locate the pairwise when significant main effects or interactions were found. The partial eta squared was converted to Cohen’s d [38] to determine the effect size (ES). The magnitude of effect size was classified as trivial (<0.20), small (0.20–0.49), medium (0.50–0.79), and large (0.80 and greater) [38]. The ES is reported in conjunction with the 95% confidence interval (CI) for all analyzed measures. SPSS version 26 for Windows (IBM Corp, Armonk, NY, USA) was used for analyses and the significance level (*p*) was set at 0.05.

## 3. Results

Normality of the data and the homogeneity of variance were confirmed. The control groups did not show any significant differences in anthropometric or athletic measures relative to the experimental groups at baseline in either circa- or post-PHV (all *p* > 0.05) (Table 1 and Table 3). There were no significant anthropometric changes in all groups after the training period (Table 1). Both the experimental (all *p* < 0.0001; ES = 0.338–1.908) and control (*p* = 0.011–0.0001; ES = 0.2–1.8) groups improved their athletic performances over the 8-week intervention (Table 3).

For sprint times, there was a main effect for condition (Table 3), with the experimental groups eliciting higher improvements than CONG over all distances (S10: ES = 0.829, 95% CI = −0.055 to −0.029; S20: ES = 0.77, 95% CI = −0.090 to −0.062; Ball-S10: ES = 0.423, 95% CI = −0.099 to −0.065; and Ball-S20: ES = 3.471, 95% CI = −0.162 to −0.120; all *p* < 0.0001) in either circa- (Δ experimental vs. Δ CONG: S10: *p* = 0.003, 95% CI = −0.73 to −3.23, ES = 1.327; S20: *p* < 0.0001, 95% CI = −1.53 to −2.68, ES = 2.737; Ball-S10: *p* < 0.0001; 95% CI = −2.35 to −4.46; ES = 2.732; and Ball-S20: *p* < 0.0001; 95% CI = −2.74 to −4.14; ES = 4.566) or post-PHV (Δ experimental vs. Δ CONG: S10: ES = 1.582, 95% CI = −1.19 to −3.69; S20: ES = 4.024, 95% CI = −1.92 to −3.07; Ball-S10: ES = 2.588, 95% CI = −1.77 to −3.54; and Ball-S20: ES = 3.374, 95% CI = −2.45 to −3.85; all *p* < 0.0001) (Table 3).

There was also a condition effect (Table 3), with significant improvements in COD-15 m and Ball-15 m times for the experimental groups relative to CONG (all *p* < 0.0001; ES = 0.604 and 0.404, 95% CI = −0.193 to −0.135, and −0.295 to −0.164, respectively) in either circa- (Δ experimental vs. Δ CONG: COD-15 m: ES = 2.766, 95% CI = −2.64 to −4.98; Ball-15 m: ES = 1.659, 95% CI = −3.61 to −6.74; all *p* < 0.0001) or post-PHV (Δ experimental vs. Δ CONG: COD-15 m: ES = 3.381, 95% CI = −2.79 to −4.96; Ball-15 m: ES = 2.709, 95% CI = −2.68 to −5.80; all *p* < 0.0001) (Table 3).

For CMJ, the training period led to a greater improvement in the experimental groups than CONG (*p* < 0.0001; 95% CI = 1.290 to 2.406; ES = 0.263) in either circa- (Δ experimental vs. Δ CONG: *p* < 0.0001, ES = 1.43, 95% CI = 3.20 to 9.99) or post-PHV (Δ experimental vs. Δ CONG: *p* < 0.0001, ES = 2.16, 95% CI = 3.99 to 10.78) (Table 3). In contrast, no main effect for the condition was identified in the dynamic balance, MAV and VO_2_max scores (Table 3). Moreover, no statistical interactions or main effects for biological age were observed in all the assessed variables (Table 3). 

## 4. Discussion

The aim of this study was to examine the in-season effects of short combined plyometric and CODB training programs (twice a week) on some measures of the athletic performances of male youth soccer players, circa- and post-PHV. After 8 weeks of intervention, the experimental condition induced significant additional gains only on explosive measures compared to the control condition, which confirms our first hypothesis in part. Our findings also showed that both the experimental and control groups improved their athletic performances over the training period. As hypothesized, circa-PHV groups did not show any significant differences in pre-to-post changes relative to the post-PHV groups.

As mentioned above, all experimental groups achieved greater improvements in their sprint, jump, and COD performances than CONG. Previous studies on youth team-sport athletes of different maturational status have highlighted improvements in power performances (i.e., S10, S20, CMJ, and COD-15 m) after different training modalities, including combined plyometric and COD drills training, PT, COD training with or without the ball, and specific circuits with the ball [1,13,14,17,35,36,39,40]. These additional meaningful adaptations in the power abilities of P-CODBT programs might be associated with neuromuscular adaptations, such as enhanced excitability of the stretch reflex system, fast nerve conduction, increased maximal voluntary contraction, and changes in the mechanical and structural characteristics of the lower limb’s muscles (e.g., muscle architecture, intramuscular coordination, and motor unit recruitment) [1,19,36,41]. Moreover, it has been reported that athletes who perform jumping and COD drills could reduce the ground contact time (especially during acceleration) and become more economical during soccer-specific motions, such as sprints, jumps, and COD activities [1,19,41]. Since horizontal-oriented force production is highly involved in the sprint performance in team sports [14], maximizing horizontal acceleration (e.g., hopping with horizontal displacement and COD exercises) seems to be an optimal conditioning stimulus for achieving power and strength improvements [14,20]. Furthermore, the additional gains in power performances could be attributed to the progressively increased training load (volume and intensity) during the experimentation. It is worth noting that prior to the intervention, our experimental groups performed only one session to train power, strength, and their derivatives (acceleration and sprinting). Interestingly, the use of selected P-CODB training exercises was effective in providing greater enhancements in the time of tests with the ball (i.e., S10-Ball, S20-Ball, and 15 m-Ball) compared with CONG. Likewise, incorporating actions that are biomechanically and metabolically specific to the performance tasks is attractive for further improving fitness performance [24], and could thus be a suitable stimulus to stress the athletic abilities of interest, as well as some technical skills (e.g., dribbling ability), in youth soccer players [13]. Indeed, the present training program extensively included COD with ball and jumping drills performed at a maximal effort (~20 min/session), which offer better results in power and strength performances than regular soccer training.

In the present study, no main effect for condition was found in the dynamic balance scores. This result contrasts previous reports that recorded additional gains in young soccer players (i.e., pre-, circa- and post-PHV) following 8 weeks of combined plyometric and COD exercise protocols [35,39,40]. The lack of difference in the dynamic balance between conditions was surprising, since prior studies have reported a good connection between dynamic balance performance and COD exercises [40,42]. In fact, rapid COD activities impose frequent perturbations upon postural control and thus, may stress the dynamic balance [42]. The reasons that can explain why the dynamic balance was not affected by additional training might be related in part to the strategy used to process postural-related information [43,44] by our regional level players. In fact, it has been demonstrated that high-level players use more short-long information (i.e., vestibular), while low-level players use more short-loop information (i.e., proprioceptive myotatic and plantar cutaneous) [44], which may consequently influence the players’ responses to learning and training stimuli. Therefore, the evidence suggests that the movement skill level should be associated with balance skill level in youth soccer players.

Although the scores for the 20 m shuttle run test increased for all groups after the intervention period, no main effect for condition was detected. The lack of significant difference between the experimental groups and CONG was not expected, since the COD exercises intensively used in the experimental condition largely mimic the motions performed during the intensive-endurance test (i.e., 20 m shuttle run test). In fact, it has been reported that the greatest transfer effects on the athletic ability of interest (i.e., VO_2_max performance) may occur when the training exercise modes (i.e., COD exercises) most closely resemble the testing procedures [45,46]. In contrast, the improved VO_2_max performance in the experimental condition, which was ~2.5% as high as that recorded for CONG, is encouraging to examine the effect of a longer duration or more frequent (>2 sessions/week) training program for youth soccer players.

A novel aspect of this study was to examine the interaction effects of the P-CODBT program and biological age on youth soccer players’ fitness. Our findings have shown no significant differences in the training-related motor adaptations between circa- and post-PHV cohorts. A meta-analysis that examined the influence of biological age on PT-related effects on COD performance in youth team-sport athletes, reported no significant differences between circa- and post-PHV groups in COD performance gains (ES for change = 0.95 vs. 0.99) [23]. In fact, it has been mentioned that the marked differences in COD performance gains between the pre- and circa-PHV stages (ES for change = 0.68 vs. 0.95) may possibly attenuate with the increased anabolic hormone concentrations and the development of the central nervous system that occur with increased age [22,23,47,48]. Furthermore, Asadi et al. [26] have shown that 6 weeks of PT (performed twice a week) induced no significant differences between circa- and post-PHV groups in motor performance gains (sprint, ES = 0.58 vs. 0.66; sprint with the ball, ES = 0.80 vs. 0.55; CMJ, ES = 0.57 vs. 0.73, respectively) in youth soccer players. In accordance with age-related athletic development, previous studies [26,49,50] reported that older youths showed a meaningful tendency toward greater explosive changes in responses to PT compared to younger youths (i.e., pre-PHV). This is partly due to various neuromuscular adaptations, such as enhanced structural and mechanical muscle properties, improved motor neuron excitability, and better quality of movement and coordination [26,51]. Regarding balance and endurance abilities, it is difficult to compare the current findings with the prior research since the influence of maturation on these abilities’ responses has not been studied after plyometric and/or COD with or without the ball protocols in both maturity statutes. Thus, further studies are warranted to elucidate the influence of maturation on balance and endurance adaptations after different training modes (e.g., PT, combined plyometric and COD with or without the ball training) in youth soccer players. This can help coaches plan proper strength and conditioning protocols to improve motor and technical skills and prevent injury. From a training prescription, the similarity in athletic responses between circa- and post-PHV may be due to the adequate stimulus of the combined plyometric and CODB program for both biological ages. In fact, both groups within this investigation had no prior experience of P-CODB training, therefore, using an equal training volume with low to moderate-high intensities may have induced comparable levels of adaptations for circa- and post-PHV cohorts.

Some limitations of this study should be stated. First, it was not possible to recruit the pre-PHV group due to logistical constraints (i.e., players’ availability at this age range), considering athletic adaptations to training stimuli can differ between these biological ages [26]. The second limitation is associated with sex and competitive level. Therefore, extending our findings to other populations (e.g., sub-elite or elite male young players, amateur or professional senior players, and female players) could be speculative. Future interventions are warranted to further understand how other populations might adapt to the P-CODBT program. Third, due to the lack of electrophysiological and biomechanical measures and muscle damage assessments, it was not possible to clarify the potential mechanisms (e.g., muscle architecture and muscle action) behind the changes observed in the power variables within age groups. Finally, there was an absence of a group performing only PT and another group performing only CODB exercises. Such a study setting would have helped to highlight which strategy is more effective.

## 5. Conclusions

The current data show that a short-term combined training protocol of plyometric and sprint with CODB exercises (2 sessions per week) can be a safe strategy that induces meaningful benefits in the explosive performances (i.e., speed with and without a ball, jump, and COD with and without a ball) of youth soccer players during the season period. Therefore, coaches and practitioners can integrate the protocols proposed in this study with their training weekly routines as part of a well-structured warm-up before specific soccer training. Since training programs should incorporate drills that closely resemble competition motions, the use of P-CODBT may be a suitable and alternative strategy to improve both the athletic and technical abilities of youth soccer players. Considering the limited available time for physical training during the season, the P-CODBT program may allow coaches to save more time for technical and tactical purposes. However, the improvements generated by P-CODBT were not affected by biological age. Further study elucidating maturational responses following P-CODBT with more maturity stages is therefore warranted.

## Figures and Tables

**Figure 1 biology-12-00120-f001:**
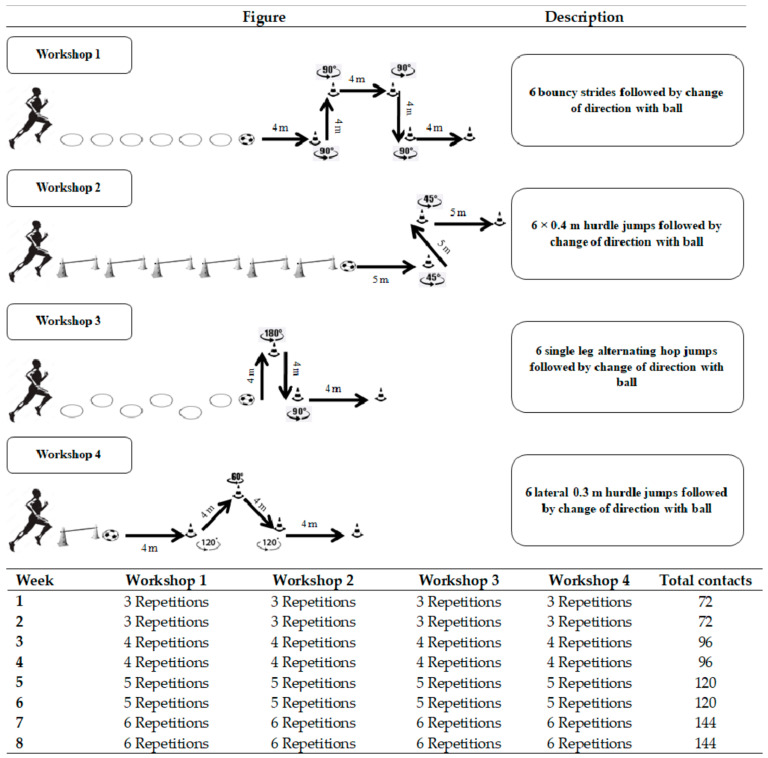
Description of the combined plyometric and sprint with change of direction with ball training program.

**Table 1 biology-12-00120-t001:** Participants’ physical characteristics (mean ± SD) (*n* = 48).

	Circa-PHV				Post-PHV			
	Experimental (*n* =12)		Control (*n* = 12)		Experimental (*n* =12)		Control (*n* = 12)	
	Pre-Test	Post-Test	Pre-Test	Post-Test	Pre-Test	Post-Test	Pre-Test	Post-Test
Age (years)	14.2 ± 0.3	14.3 ± 0.2	14.1 ± 0.2	14.3 ± 0.3	16.3 ± 0.2	16.4 ± 0.2	16.2 ± 0.3	16.3 ± 0.2
Height (cm)	164.4 ± 4.9	165.6 ± 4.9	163.6 ± 6.8	164.9 ± 6.5	176.2 ± 5.7	177.1 ± 5.5	174.4 ± 3.6	175.6 ± 3.7
LL (cm)	85.1 ± 2.4	85.5 ± 2.3	84.0 ± 3.4	84.7 ± 3.4	90.0 ± 2.8	90.6 ± 2.7	89.3 ± 2.4	89.8 ± 2.2
SH (cm)	79.1 ± 2.6	80.2 ± 2.9	79.6 ± 3.7	80.3 ± 3.3	86.2 ± 3.2	86.5 ± 2.9	85.2 ± 1.6	85.8 ± 1.7
Weight (kg)	48.3 ± 4.5	49.1 ± 4.5	50.1 ± 8.3	49.9 ± 7.6	65.0 ± 6.4	65.5 ± 5.3	64.6 ± 4.9	65.3 ± 5.5
BMI (kg·m^−2^)	17.6 ± 1.3	17.9 ± 1.3	19.3 ± 2.4	18.3 ± 2.4	21.0 ± 2.6	21.0 ± 2.2	21.3 ± 1.8	21.2 ± 2.2
APHV (years)	14.8 ± 0.4	-	14.7 ± 0.4	-	14.9 ± 0.5	-	15.0 ± 0.3	-
Predicted years from APHV	−0.6 ± 0.3	-	−0.6 ± 0.5	-	1.4 ± 0.4	-	1.2 ± 0.2	-

SD: standard deviation; LL: leg length; SH: sitting height; BMI: body mass index; APHV: age of peak height velocity.

**Table 2 biology-12-00120-t002:** Weekly training routine of youth soccer players during intervention.

Week Days	Objectives	Time	Duration (min)
Monday	Day off: physical and mental recovery	-	-
Tuesday	Aerobic training and technical–tactical drills	18:00–20:00	85–90
Wednesday	Small-sided games and technical–tactical drills	18:00–20:00	85–90
Thursday	Power anaerobic training and technical–tactical drills	18:00–20:00	85–90
Friday	Speed training, technical drills, and simulated competitive games	18:00–20:00	85–90
Saturday	Reaction speed and technical–tactical drills	14:00–15:30	60–70
Sunday	Official match	-	-

**Table 3 biology-12-00120-t003:** Athletic measures at baseline and after the intervention period for all groups (mean ± SD) (*n* = 48).

		Experimental				Control				ANCOVA
		Pre-Test	Post-Test	ES (*p* Value), 95 %CI	Δ	Pre-Test	Post-Test	ES (*p* Value), 95 %CI	Δ	
S10 (s)	Circa-PHV	1.73 ± 0.10	1.64 ± 0.09 †	0.946 (<0.0001), 0.07–0.10	−4.94 ± 1.42 ‡	1.77 ± 0.09	1.71 ± 0.09 †	0.667 (<0.0001), 0.03–0.07	−2.96 ± 1.56	Condition: F = 40.407; *p* < 0.0001; ES = 0.935 Maturation: F = 0.567; *p* = 0.455; ES = 0 Interaction: F = 0.085; *p* = 0.722; ES = 0
	Post-PHV	1.70 ± 0.07	1.62 ± 0.06 †	1.227 (<0.0001), 0.06–0.09	−4.58 ± 1.45 ‡	1.72 ± 0.09	1.68 ± 0.08 †	0.532 (=0.0005), 0.02–0.05	−2.14 ± 1.63	
S20 (s)	Circa-PHV	3.32 ± 0.18	3.18 ± 0.17 †	0.8 (<0.0001), 0.12–0.16	−4.31 ± 0.91 ‡	3.39 ± 0.17	3.32 ± 0.16 †	0.424 (<0.0001), 0.06–0.09	−2.20 ± 0.73	Condition: F = 121.457; *p* < 0.0001; ES = 1.636 Maturation: F = 0.645; *p* = 0.426; ES = 0 Interaction: F = 0.672; *p* = 0.417; ES = 0
	Post-PHV	3.26 ± 0.14	3.10 ± 0.14 †	1.143 (<0.0001), 0.14–0.16	−4.69 ± 0.58 ‡	3.29 ± 0.17	3.22 ± 0.17 †	0.412 (<0.0001), 0.06–0.08	−2.19 ± 0.66	
Ball-S10 (s)	Circa-PHV	2.36 ± 0.32	2.22 ± 0.29 †	0.458 (<0.0001), 0.11–0.17	−5.84 ± 1.37 ‡	2.50 ± 0.27	2.44 ± 0.25 †	0.231(<0.0001), 0.04–0.07	−2.24 ± 0.82	Condition: F = 95.003; *p* < 0.0001; ES = 1.445 Maturation: F = 0.392; *p* = 0.534; ES = 0 Interaction: F = 0.778; *p* = 0.384; ES = 0
	Post-PHV	2.24 ± 0.34	2.10 ± 0.33 †	0.418 (<0.0001), 0.10–0.15	−5.89 ± 1.53 ‡	2.19 ± 0.25	2.14 ± 0.25 †	0.2 (<0.0001), 0.04–0.07	−2.48 ± 0.88	
Ball-S20 (s)	Circa-PHV	4.46 ± 0.62	4.21 ± 0.59 †	0.413 (<0.0001), 0.22–0.28	−5.64 ± 0.91 ‡	4.59 ± 0.35	4.49 ± 0.35 †	0.286 (<0.0001), 0.08–0.012	−2.19 ±0.56	Condition: F = 180.211; *p* < 0.0001; ES = 1.995 Maturation: F = 1.820; *p* = 0.184; ES = 0.134 Interaction: F = 1.579; *p* = 0.216; ES = 0.113
	Post-PHV	4.10 ± 0.54	3.89 ± 0.53 †	0.393 (<0.0001), 0.18–0.24	−5.19 ± 1.10 ‡	4.14 ± 0.43	4.05 ± 0.42 †	0.212 (<0.0001), 0.06–0.10	−2.04 ± 0.73	
COD−15 m (s)	Circa-PHV	4.34 ± 0.47	4.09 ± 0.43 †	0.555 (<0.0001), 0.20–0.29	−5.68 ± 1.14 ‡	4.50 ± 0.43	4.41 ± 0.39 †	0.219 (=0.004), 0.03–0.14	−1.87 ±1.58	Condition: F = 127.510; *p* < 0.0001; ES = 1.676 Maturation: F = 3.757; *p* = 0.059; ES = 0.247 Interaction: F = 1.048; *p* = 0.312; ES = 0.032
	Post-PHV	3.83 ± 0.26	3.60 ± 0.25 †	0.902 (<0.0001), 0.19–0.27	−6.02 ± 1.50 ‡	3.89 ± 0.27	3.81 ± 0.27 †	0.296 (<0.0001), 0.07–0.10	−2.14 ± 0.62	
Ball−15 m (s)	Circa-PHV	5.60 ± 0.62	5.21 ± 0.56 †	0.66 (<0.0001), 0.28–0.49	−6.86 ± 2.57 ‡	5.76 ± 0.85	5.60 ± 0.75 †	0.2 (=0.0112), 0.04–0.26	−2.48 ±2.71	Condition: F = 50.174; *p* < 0.0001; ES = 1.045 Maturation: F = 2.877; *p* = 0.097; ES = 0.204 Interaction: F = 0.557; *p* = 0.459; ES = 0.013
	Post-PHV	5.18 ± 0.42	4.81 ± 0.38 †	0.924 (<0.0001), 0.31–0.43	−7.11 ± 1.69 ‡	5.08 ± 0.63	4.93 ± 0.55 †	0.254 (=0.0003), 0.09–0.21	−2.87 ± 1.43	
CMJ (cm)	Circa-PHV	24.6 ± 2.9	27.7 ± 3.0 †	1.051 (<0.0001), 2.30–3.94	12.9 ± 5.9 ‡	26.5 ± 3.4	28.2 ± 3.7 †	0.478 (<0.0001), 1.17–2.19	6.3 ±2.8	Condition: F = 44.619; *p* < 0.0001; ES = 0.984 Maturation: F = 2.555; *p* = 0.117; ES = 0.185 Interaction: F = 3.011; *p* = 0.090; ES = 0.211
	Post-PHV	31.9 ± 4.0	35.8 ± 3.5 †	1.038 (<0.0001), 2.45–4.40	12.7 ± 4.0 ‡	31.1 ± 3.6	32.8 ± 3.7 †	0.466 (<0.0001), 1.06–2.17	5.3 ± 3.1	
Balance scores (%)	Circa-PHV	101.2 ± 5.4	104.4 ± 5.2 †	0.604 (<0.0001), 2.40–3.93	3.2 ± 1.2	99.8 ± 9.2	102.1 ± 9.1 †	0.251 (<0.0001), 1.83–2.84	2.4 ±0.9	Condition: F = 3.833; *p* = 0.057; ES = 0.250 Maturation: F = 0.204; *p* = 0.654; ES = 0 Interaction: F = 0.025; *p* = 0.875; ES = 0
	Post-PHV	90.6 ± 5.3	93.7 ± 5.0 †	0.602 (<0.0001), 2.30–3.91	3.5 ± 1.5	91.7 ± 4.3	94.1 ± 4.9 †	0.521 (=0.0004), 1.39–3.51	2.7 ± 1.8	
MAV (km·h^−1^)	Circa-PHV	10.9 ± 0.8	12.0 ± 0.8 †	1.375 (<0.0001), 0.93–1.32	10.4 ± 2.9	10.6 ± 0.5	11.5 ± 0.5 †	1.8 (<0.0001), 0.73–1.10	8.7 ± 2.8	Condition: F = 3.648; *p* = 0.063; ES = 0.242 Maturation: F = 0.646; *p* = 0.426; ES = 0 Interaction: F = 0.226; *p* = 0.637; ES = 0
	Post-PHV	12.2 ± 0.6	13.4 ± 0.7 †	1.841 (<0.0001), 1.08–1.41	10.3 ± 2.1	12.0 ± 1.0	13.0 ± 0.9 †	1.0 (<0.0001), 0.67–1.24	8.5 ± 4.2	
VO_2_max (mL·min^−1^·kg^−1^)	Circa-PHV	43.8 ± 4.3	49.8 ± 4.4 †	1.379 (<0.0001), 4.86–7.01	13.7 ± 4.1	42.4 ± 3.1	47.1 ± 3.0 †	1.541 (<0.0001), 3.60–5.83	11.3 ± 4.3	Condition: F = 3.940; *p* = 0.054; ES = 0.255 Maturation: F = 0.085; *p* = 0.773; ES = 0 Interaction: F = 0.239; *p* = 0.627; ES = 0
	Post-PHV	47.9 ± 3.3	55.0 ± 4.1 †	1.908 (<0.0001), 6.12–8.03	14.8 ± 3.0	47.1 ± 5.9	52.7 ± 5.4 †	0.914 (<0.0001), 3.61–7.05	12.3 ± 6.9	

SD: standard deviation; S10: 10 m sprint; S20: 20 m sprint; Ball-S10: 10 m sprint with ball; Ball-S20: 20 m sprint with ball; COD−15 m: 15 m agility run without ball; Ball−15 m: 15 m agility run with ball; CMJ: countermovement jump; MAV: maximal aerobic velocity; VO_2_max: maximal oxygen consumption. ES: effect size; 95%CI: 95% confidence interval. †: A significant difference when comparing T1 and T2; ‡: Significantly different from Δ control.

## Data Availability

The data presented in this study are available on request from the corresponding author. The data are not publicly available due to privacy reasons.

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
