# Peer review of "Effects of Biological Age on Athletic Adaptations to Combined Plyometric and Sprint with Change of Direction with Ball Training in Youth Soccer Players"

_biology, 2023, doi:10.3390/biology12010120_

Round 1
Reviewer 1 Report
The manuscript is focused on an interesting topic, the introduction of a plyometric + change of direction training on several soccer-related skills in circa-and post-PHV young players. The data showed that the experimental condition induced significant additional gains only on explosive measures compared to the control condition (i.e., speed with and without ball, jump, COD with and without ball) in youth soccer players during the season period.
The ms has some potentialities, still some points to be clarified remains:
1) Introduction. This section is too long. I suggest the Authors to be more focused and shorten the Introduction. Moreover, on which bases do the Authors hypothesised that biological differences will have no effects?
2) Methods. This part is complete and well-written. I have only some perplexity about the statistical approach. Would a mixed-general model ANCOVA with time as repeated, group as between-group effect and biological-age as covariate be a more appropriate approach that repeated T-tests?
3) Results. Clear and well reported.
4) Discussion. Clear
Author Response
Referee 1
Comment 1
1) Introduction. This section is too long. I suggest the Authors to be more focused and shorten the Introduction. Moreover, on which bases do the Authors hypothesised that biological differences will have no effects?
We thank the expert referee for his/her comment.
- We have tried to shorten the introduction. Please find changes in the text.
- We hypothesized that biological differences will have no effects on athletic performances based on the results of Asadi et al (27) who have shown that 6 weeks of PT (performed twice a week) induced no significant differences between circa- and post-PHV groups in motor performance gains (sprint, ES = 0.58 vs. 0.66; sprint with ball, ES = 0.80 vs. 0.55; CMJ, ES = 0.57 vs. 0.73, respectively) in youth soccer players.
Comment 2
Methods. This part is complete and well-written. I have only some perplexity about the statistical approach. Would a mixed-general model ANCOVA with time as repeated, group as between-group effect and biological-age as covariate be a more appropriate approach that repeated T-tests?
Author's response:
We thank very much the expert referee for his/her comment.
‘’Independent samples t-test was performed to examine between-group differences at pre-test. Regarding pre-test differences among maturity groups, ANCOVAs (Biological age [circa-PHV and post-PHV] × Condition [Experimental and CONG]) with pre-test as a covariate were performed to test biological age and condition effects on the dependent variable values. However, we used a paired t-test to examine the effectiveness of the training program within all groups.’’
Reviewer 2 Report
The scientific article is very interesting and has a great difficulty in being able to carry out a sports teams practical intervention in sports season. The objective of the study is perfectly detailed in contrasting the effectiveness of a type of training in young people and its impact on growth.
The scientific methodology and documentary sources are appropriate to the study, meeting the requirements required and demanded by the scientific journal to which it is submitted.
There are several considerations that this review wants to be assessed and specified or modified within the final text of the article.
In the first place when it indicates line 92 “The SSC is of great importance in the PT due to its participation in improving the capacity of the muscle-tendon unit and in the production of maximum force in the shortest possible time [20]. ” Why don't you use the term RFD to measure this application of the maximum force in the shortest possible time?
In line 97 it indicates "efficacy and safety of combined plyometric and sprint training with change of direction in young people [1,17]" but it does not mention at any time about athlete's health, these methods have been classified as not recommended for athletes young people precisely because of this lack of basic muscular preparation in young people in sports construction like the CIRCA group.
In line 138, regarding the determination of the PHV, the reviewer wonders if the applied methodology is really adequate for this publication, since currently the most scientifically recognized Kinanthropometric indicators are used in peer-reviewed publications. Can you better argue your choice in the article?
In line 233, it uses suitable instruments for measuring the CMJ Test, but going back to the reflection that I have made previously, the design would have been more complete and scientifically recognized with the use of force platforms and seeking the application of variables such as RFD. In several recognized scientific journals, this instrument (force platform) and variable (RFD) is necessary to contrast the hypotheses of the work they present.
In line 434 I agree with you on the limitations of the study, including the proper biomechanical determination to determine the possible strength gains.
I look forward to your contributions to the points raised.
Author Response
Referee 2
Comments 1 and 4
- In the first place when it indicates line 92 “The SSC is of great importance in the PT due to its participation in improving the capacity of the muscle-tendon unit and in the production of maximum force in the shortest possible time [20]. ” Why don't you use the term RFD to measure this application of the maximum force in the shortest possible time?
- In line 233, it uses suitable instruments for measuring the CMJ Test, but going back to the reflection that I have made previously, the design would have been more complete and scientifically recognized with the use of force platforms and seeking the application of variables such as RFD. In several recognized scientific journals, this instrument (force platform) and variable (RFD) is necessary to contrast the hypotheses of the work they present.
Author's response:
Thank you for your valuable comment.
As you said, RFD outcomes have been considered to have important functional consequences because of their temporal similarity with respect to sport and daily activities and because of their positive correlation with performances in both sporting and functional daily tasks. Unfortunately, we do not have the necessary instrument (force platform) to measure this variable. However, we have mentioned as a limitation that the lack of electrophysiological and biomechanical measures makes it difficult to clarify the potential mechanisms (e.g., muscle architecture, muscle action) behind the changes observed in the power variables within age groups.
Comment 2
- In line 97 it indicates "efficacy and safety of combined plyometric and sprint training with change of direction in young people [1,17]" but it does not mention at any time about athlete's health, these methods have been classified as not recommended for athletes young people precisely because of this lack of basic muscular preparation in young people in sports construction like the CIRCA group.
Author's response:
Thank you for your valuable comment.
The word “safety” has been deleted. Please find changes in the text.
Comment 3
- In line 138, regarding the determination of the PHV, the reviewer wonders if the applied methodology is really adequate for this publication, since currently the most scientifically recognized Kinanthropometric indicators are used in peer-reviewed publications. Can you better argue your choice in the article?
Author's response:
We thank the expert referee for his/her comment.
The influence of biological maturity status (BMS) on talent identification and development within elite youth soccer is critically debated (Leyhr et al., 2020). While standard methods for assessing BMS in adolescents are expensive and time-consuming imaging techniques [i.e., X-ray and MRI (skeletal age)], there also exist more pragmatic procedures (maturity offset, percentage of adult height) (Mirwald et al., 2002; Khamis and Roche, 1994). For instance, Mirwald's equation (Mirwald et al., 2002) calculates both BMS and maturity timing (Leyhr et al., 2020). The results of prior studies (Leyhr et al., 2020) suggested that the use of maturity offset and percentage of adult height for measuring BMS is more pragmatic in terms of cost and time as compared with MRI diagnostics. Based on a general agreement between these pragmatic diagnostics and the reference method MRI in all three perspectives, the alternative methods can be used to determine BMS among (male) elite youth soccer players (Leyhr et al., 2020). Since caution is required with respect to the precision of the measurements at the individual level, the simultaneous use of at least two alternative diagnostics is recommended in order to get a more reliable BMS outcome (Leyhr et al., 2020).
References:
Khamis, H., and Roche, A. (1994). Predicting adult stature without using skeletal age: the Khamis-Roche method. Pediatrics 94, 504–507.
Leyhr D, Murr D, Basten L, Eichler K, Hauser T, Lüdin D, Romann M, Sardo G, Höner O. (2020). Biological Maturity Status in Elite Youth Soccer Players: A Comparison of Pragmatic Diagnostics With Magnetic Resonance Imaging. Front Sports Act Living 2, 587861.
Mirwald, R., Baxter-Jones, A., Bailey, D., and Beunen, G. (2002). An assessment of maturity from anthropometric measurements. Med. Sci. Sports Exerc. 34, 689–694.